# The Essential Oil Composition of *Trachymene incisa* Rudge subsp. *incisa* Rudge from Australia [note 1]

**DOI:** 10.3390/plants10030601

**Published:** 2021-03-23

**Authors:** Jesús Palá-Paúl, Lachlan M. Copeland, Joseph J. Brophy

**Affiliations:** 1Departamento Biodiversidad Ecología y Evolución, Facultad de Biología, Universidad Complutense de Madrid, 28040 Madrid, Spain; 2Eco Logical Australia, 24 Gordon St, Coffs Harbour, NSW 2450, Australia; Lachlanc@ecoaus.com.au; 3School of Chemistry, The University of New South Wales, Sydney, NSW 2052, Australia; J.Brophy@unsw.edu.au

**Keywords:** *Trachymene incisa*, Apiaceae, chemotype, β-selinene, bicyclogermacrene, γ-bisabolene, α-pinene, β-caryophyllene, essential oil, chemical composition

## Abstract

*Trachymene incisa* subsp. *incisa* is an Australian endemic taxon that varies greatly in the abundance and length of the leaf trichomes. The essential oil composition of five populations of this subspecies, three corresponding to the typical glabrous form and two of the particularly hairy variant, has been analyzed in an attempt to determinate if that variability is also reflected in their composition. The oils have been extracted by hydrodistillation and analyzed by Gas Chromatography (GC) and Gas Chromatography coupled to Mass Spectrometry (GC–MS). The essential oils of *T. incisa* subsp. *incisa* were characterized by the high amount of sesquiterpenes that were the major fraction. The sesquiterepene hydrocarbons were significantly higher in the hairy variant in comparison to the glabrous one. According to the main compound, three different chemotypes were found: I.—β-selinene + bicyclogermacrene and II.—γ-bisabolene + α-pinene for the typical glabrous variant and III.—bicyclogermacrene + β-caryophyllene for the hairy variant.

## 1. Introduction

The genus *Trachymene* Rudge belongs to the Apiaceae family and comprises at least 56 species of herbs, distributed in Australia, New Guinea, the Philippine Islands, Borneo, New Caledonia, Vanuatu and Fiji. Of those, 38 species are native to Australia with a wide diversity of habitats ranging from deserts to alpine areas. The taxonomy of this genus has been controversial throughout history because of the numerous synonyms and the plasticity of some of the species [1]. The type section, *Trachymene* sect. *Trachymene*, includes perennial species with functionally bicarpellate fruits [2].

*Trachymene incisa* Rudge subsp. *incisa* (Figure 1) is an erect or ascending perennial herb up to 80 cm high with a thick rootstock. The plants have highly variable dissected basal leaves and one or more rosettes attached to the rootstock. The leaves are sparsely hairy to almost glabrous, the lamina broad ovate, 14–65 mm long, 15–76 mm wide; segments elliptic or linear, 0.5–6 mm wide. Plants have bisexual white flowers, clustered into simple umbels with a mean of 72 flowers per umbel. The fruits are broad ovate and are 2–4 mm long, 2.5–4.0(–7.5) mm wide, brown; occasionally developing only one mericarp. Mericarps are 1.8–2.0(–3.8) mm wide; smooth orbiberculate to papillate schizocarps. It flowers most of the year, peaking between September and April. The dispersal takes place from March to October with no obvious peak. It grows commonly in dry eucalypt woodland or scrub, sclerophyll forest and cleared areas on sandy soils and on rocky granite outcrops and resprouts from the tuberous taproot after fire. It is distributed widely throughout eastern NSW north from Ulladulla and into Queensland [1,3,4,5,6,7,8,9]. 

Research on this genus has focused on medicinal or ecological aspects. In southwest Queensland and northwest New South Wales, different species of *Trachymene* (*T. glaucifolia* (F. Muell.) Benth., *T. cyanantha* Boyland and *T. ochracea* L.A.S. Johnson) seem to be toxic to cattle [10,11,12,13,14]. The only herbarium voucher of them corresponded to *T. glaucifolia* that seems to be directly correlated to a form of “staggers” in sheep and the deformity “bent leg” in lambs [13]. Further research has correlated the intake of *T. ochracea* with the limb paresis syndrome, included in the first level of the five progressive clinical groups described of the nervous and muscular locomotor disorders that affect sheep throughout Australia [14]. The genus *Trachymene* has been also reported to show teratogenic compounds that affect the development of an embryo in pregnancy or may cause a birth defect in the child [15]. On the other hand, the niche and competitive ability between *T. cyanopetala* (F. Muell.) Benth. and *T. ornata* (Endl.) Druce was evaluated, although little evidence was found under well-watered and water-stressed conditions [16]. *Phytophthora caetorum* (Lebert and Cohn) Schroeter is a pathogen fungus that attacks *T. caerulea* R. Grah., the blue lace flower that is field-grown for the cut-flower industry, although the use of hymexazol (35 mg/L) reduced significantly the symptoms [17]. Another species, *T. pilosa* Sm., has been described as a resistant and key symptomless host of a relative pathogen *P*. *cinnamomi* Rands. However, under laboratory conditions, this species died [18,19,20]. The resistance of the exine from the mature pollen grains of this species has been estimated [21]. The regeneration ability of *T. coerulea* Graham under experimental conditions showed that the center of the shoot apex produced fewer mature leaves and fewer embryos from the basal tissues than did the flanks [22]. Finally, the importance of this genus is clear when species such as *T. cussonii* (Montrouz.) B.L. Burtt have been described as a coral cay endemic plant representative to the conservation of this type of habitat [23,24].

This species has been previously reported as being a member of the Woodland endangered ecological community [25]. However, most of the papers focus on its reproductive ecology, pollination and germination [4,7,26,27].

The aim of this research was to contribute to the knowledge of the essential oil composition of *Trachymene incisa* subsp. *incisa* and evaluate if the abundance of trichomes can affect the composition. Apart from our preliminary conference report, as far as we know this is the first report about the chemical composition of the essential oil of this species, or the genus.

## 2. Results and Discussion

The distillation of the aerial parts of five different populations of *T. incisa* subsp. *incisa* yielded small amounts of a pale yellow oil (0.05–0.17%). The amount of oil obtained was reduced in comparison with other members of the Apiaceae that are well known and used as condiment or spices [28,29]. However, it would be of interest to check other species of this genus to contrast this.

The identified constituents from the aerial parts of *T. incisa* subsp. *incisa*, their retention indices and their percentage composition are summarized in Table 1, where all the compounds are arranged in order of their elution from the DB-Wax column. The chromatograms of each analyzed sample and their peak identification and included in the Appendix A. From all the samples analyzed, 47 compounds have been identified, representing from 88.4% to 93.5% of the total oil. The sesquiterpene fraction predominates, both quantitatively and qualitatively, in all the samples examined, with a total of 33 constituents. Sesquiterpene hydrocarbons (48.3–74.8%) were more abundant than oxygenated sesquiterpene (6.59–36.2%). It is worth nothing the low amount of monoterpene fraction and practically the absence of the oxygenated ones, that only appear in T.in.I_3_ (9.0%). The synthesis of terpenoids is correlated from the lower molecular weight to the higher [30,31], so this species has increased the metabolism that produces sesquiterpenes using monoterpenes as precursors. That could explain the low amount of monoterpenes detected. However, it should be interesting to make a seasonal study of the population 3 (T.in.I_3_) to know if the amount of α-pinene is constant throughout the cycle of the plant or if it could be conditioned by other factors.

According to our results, the abundance of trichomes seems to affect the composition of the essential oils of *Trachymene incisa* subsp. *incisa* (Table 1). Although the terpenoid distribution seems similar between both glabrous and hairy variants, the amount of sesquiterpene hydrocarbons is significantly higher in the hairy one (Figure 2). Only five populations were analyzed and no climatic or biological conditions were registered. Thus, further study should be done to confirm this or to check if the age of the plant, the presence of pathogen or the climatic condition could affect the chemical composition. Previous reports have confirmed that the phenology of the plant and the growing conditions can affect the essential oils of other species of the same family [32,33,34]. It has been noted that the density of seeds affects the germination of this species [7], so the competition between growth in the plants could also alter its chemistry as allelopathic compounds do [35,36,37]. In fact, the competition has been reported to have an effect on the phylogenetic signal and phenotypic plasticity in plant functional traits and to the root biomass [38,39]. More work is required to check under laboratory conditions if any of these aspects are involved in the abundance and length of the leaf trichomes of this subspecies.

With respect to the chemical composition of the volatile oils, *T. incisa* subsp. *incisa* exhibits the same plasticity in its vegetative and reproductive forms. The typical glabrous variant (T.in.I), showed β-selinene (11.9–36.7%) and bicyclogermacrene (21.5–28.0%) as main compounds in populations 1 and 2. However, the principal components of the hairy variant (T.in.II) were identified as bicyclogermacrene (24.4–34.7%) and β-caryophyllene (10.4–10.8%) in populations 4 and 5, although population 5 also contained a high amount of caryophyllene oxide (12.8%). According to these results, it seems that each variant has a characteristic composition, but population 3 belonging to the typical glabrous variant (T.in.I) showed other chemotype, with γ-bisabolene (27.4%) and α-pinene (19.2%) being the major compounds. According to the main compound, three different chemotypes could be defined: I.—β-selinene + bicyclogermacrene and II.—γ-bisabolene + α-pinene for the typical glabrous variant and III.—bicyclogermacrene + β-caryophyllene for the hairy variant, though more sampling of the species would be needed to confirm this.

With this paper, we contribute for the first time to the knowledge of the chemistry of the genus *Trachymene*. It would be interesting to study the effects of climate, light or soil conditions on the chemical composition of this species to confirm the chemotypes described. The diversity of this genus could also be the source of further investigations to characterize their essential oils.

## 3. Materials and Methods

### 3.1. Plant Material

Five different populations of *Trachymene incisa* subsp. *incisa* were gathered at flowering from the Northern Tablelands of New South Wales in 2007 (Table 2). Voucher specimens were lodged at the N.C.W. Beadle Herbarium of the University of New England, Australia (NE).

### 3.2. Isolation of Volatile Oils

The oils were isolated by steam distillation with cohobation for 8 h as previously described by Brophy et al. [40]. The oils were colorless to pale yellow, with a yield from 0.05% to 0.17% based in dry weight (Table 2).

### 3.3. Identification of Components

Analytical gas chromatography (GC) was carried out on a Shimadzu GC17A gas chromatograph with a Megabore column of DB-Wax (60 m × 0.5 mm × 1 µm) which was programmed from 50 to 220 °C at 3 °C min^−1^ with helium as the carrier gas. Injection temperature and detector temperature were both 220 °C. GC integrations were performed on a SMAD electronic integrator. GC–MS was carried out on a Shimadzu GCMS-QP5000 mass spectrometer operating at 70 eV ionization energy. The GC column used was a DB-wax (30 m × 0.25 mm × 0.25 µm) programmed from 35 to 220 °C at 3 °C min^−1^ with helium as the carrier gas, injection temperature and ion source temperature were both 250 °C. Compounds were identified by their GC retention indices relative to n-alkanes and known compounds and by comparison of their full-scan mass spectra with either known compounds or published spectra [41,42,43,44,45,46,47,48]. The oil sample was also analyzed on the same GC–MS system and under the same conditions as above except that a DB-5 (30 m × 0.25 mm × 0.25 µm) column was used, programmed from 35 to 250 °C at 5 °C min^−1^ with helium as the carrier gas. In all cases, the mass spectrometer was scanned from *m*/*z* 41 to *m*/*z* 450 amu in 1 s.

## 4. Conclusions

According to our results detailed above, the essential oils of *Trachymene incisa* subsp. *incisa* are characterized by a high amount of sesquiterpenes that were the major compounds. The sesquiterepene hydrocarbons were significantly more abundant in the hairy variant in comparison to glabrous form. It may be that three different chemotypes occur, with the typical glabrous variant showing two different chemotypes, characterized by: β-selinene + bicyclogermacrene and γ-bisabolene + α-pinene, while the hairy variant was characterized by bicyclogermacrene + β-caryophyllene.

## Figures and Tables

**Figure 1 plants-10-00601-f001:**
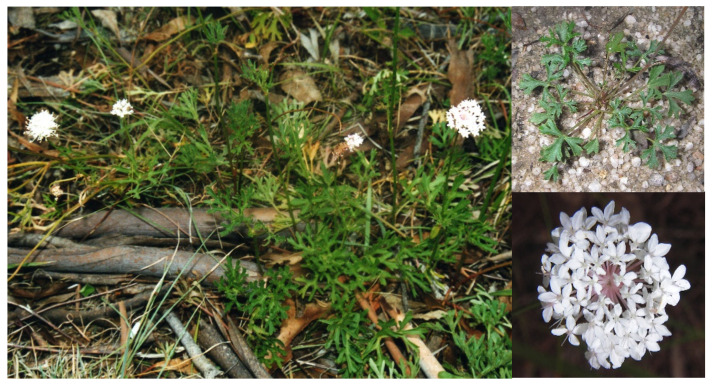
*Trachymene incisa* Rudge subsp. *incisa*. Details of the basal leaves and umbel.

**Figure 2 plants-10-00601-f002:**
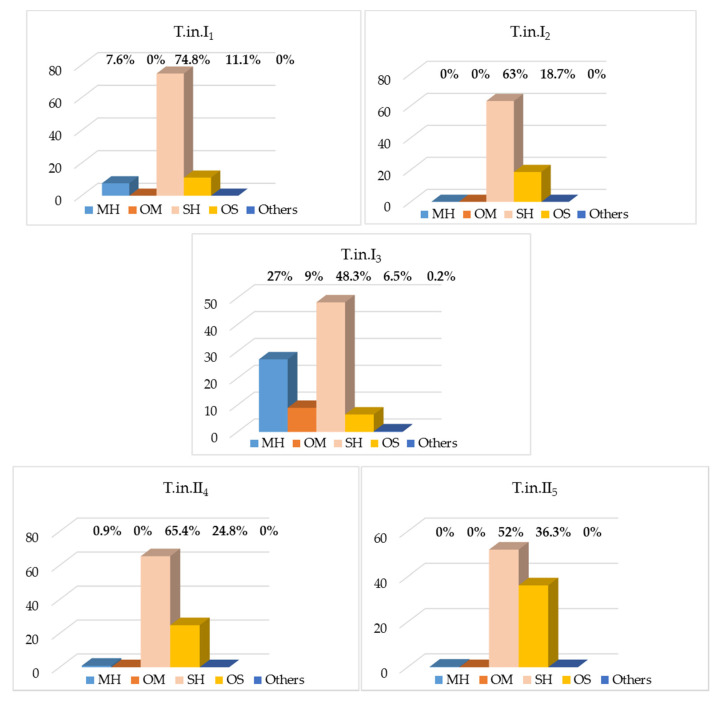
Distribution of the compound type of the essential oils of *T. incisa* subsp. *incisa* (T.In. = *Trachymene incisa* subsp. *Incisa*; I and II = typical glabrous and hairy variant, respectively).

**Table 1 plants-10-00601-t001:** Essential oil composition (%) of *Trachymene incisa* subsp. *incisa* variants.

KIa.	KI^1^	KIb	KI^2^	Compound	C. Type	T.in.I_1_	T.in.I_2_	T.in.I_3_	T.in.II_4_	T.in.II_5_	I.M.
1007	1025	932	936	α-pinene	MH	7.2		**19.2**	t	0.1	KI, MS
1034	1061	945	949	α-fenchene	MH			0.8			KI, MS
1041	1068	946	950	camphene	MH			t			KI, MS
1084	1110	974	978	β-pinene	MH	0.4		2.2	t	0.1	KI, MS
1105	1122	969	973	sabinene	MH						KI, MS
1127	1146	1008	1011	δ-3-carene	MH			1.8			KI, MS
1139	1167	1002	1004	α-phellandrene	MH			1.2			KI, MS
1141	1160	988	989	myrcene	MH	t					KI, MS
1157	1177	1014	1017	α-terpinene	MH						KI, MS
1176	1198	1024	1029	limonene	MH	t		0.8			KI, MS
1178	1209	1025	1030	β-phellandrene	MH			t			KI, MS
1181	1211	1026	1031	1,8-cineole	OM	t		9.0			KI, MS
1218	1234	1032	1038	*Z*-β-ocimene	MH			t			KI, MS
1224	1245	1054	1059	γ-terpinene	MH			t			KI, MS
1226	1250	1044	1048	*E*-β-ocimene	MH			t			KI, MS
1249	1270	1020	1024	*p*-cymene	MH	t	t	0.6	t	t	KI, MS
1260	1282	1086	1086	terpinolene	MH		t	0.4	0.9		KI, MS
1320	1237	989	986	6-methylhept-5-en-2-one	O		t	0.2	t	t	KI, MS
1459	1469	1335	1337	δ-elemene	SH	0.3	0.3		t		KI, MS
1462	1491	1374	1376	α-copaene	SH	t	3.4		t		KI, MS
1510	1541	1387	1387	β-cubebene	SH		0.6		t		KI, MS
1523	1559	1411	1414	α-*cis*-bergamotene	SH			0.6	0.4		KI, MS
1543	1575	1432	1435	α-*trans*-bergamotene	SH		0.2	2.1	t		KI, MS
1560	1590	1389	1390	β-elemene	SH	0.8	2.9	0.4	t		KI, MS
1563	1598	1417	1420	β-caryophyllene	SH	2.5	5.4	1.2	**10.8**	**10.4**	KI, MS
1570	1620	1439	1441	aromadendrene	SH	1.5	0.6	0.5	5.6	5.0	KI, MS
1572	1629	1509	1504	α-bulnesene	SH	0.5		t	1.2	0.9	KI, MS
1632	1649	1458	1460	*allo*-aromadendrene	SH	0.2	0.4	0.4	0.4	0.4	KI, MS
1644	1664	1454	1456	(*E*)-β-farnesene	SH		0.3	1.4	0.6		KI, MS
1653	1666	1452	1453	α-humulene	SH	0.5	1.5	0.3	1.8	1.8	KI, MS
1658				C_15_H_24_	SH	0.4	0.3	0.7			KI, MS
1661	1696	1496	1492	viridiflorene	SH	1.9	0.8	0.6	5.4	4.7	KI, MS
1670	1708	1484	1480	germacrene-D	SH	1.4	2.4	1.5	0.5		KI, MS
1680	1705	1505	1512	γ-bisabolene	SH			**27.4**			KI, MS
1697	1717	1489	1486	β-selinene	SH	**36.7**	**11.9**	0.5	1.8		KI, MS
1706	1734	1500	1494	bicyclogermacrene	SH	**28.0**	**21.5**	7.5	**34.7**	**24.4**	KI, MS
1722	1763	1513	1513	γ-cadinene	SH	t					KI, MS
1724	1755	1522	1521	δ-cadinene	SH	t	9.0	0.5	1.5	1.7	KI, MS
1727	1788	1495	1531	cis-cadina-1,4-diene	SH		0.3		t		KI, MS
1734	1744	1505	1504	(E,E)-α-farnesene	SH	t	1.4	2.6		1.6	KI, MS
1749	1773	1479	1482	ar-curcumene	SH		t	t	0.7	1.1	KI, MS
1886	1986	1582	1580	caryophyllene oxide	OS	1.6	2.3	0.2	1.0	**12.8**	KI, MS
1958	2039	1602	1582	ledol	OS	0.2	0.2	0.1	0.5	t	KI, MS
2016	2067	1645	1636	cubenol	OS	t	0.9	0.2	0.8	1.5	KI, MS
2018	2074	1595	1588	cubeban-11-ol	OS	0.7	0.4	0.4	1.6	1.1	KI, MS
2034	2082	1590	1582	globulol	OS	2.0	1.5	1.3	7.0	6.9	KI, MS
2044	2090	1592	1591	viridiflorol	OS	1.5	1.1	0.8	3.9	3.5	KI, MS
2066				C_15_H_26_O	OS	0.5	0.3	0.3	2.2	2.0	KI, MS
2074				C_15_H_26_O	OS	0.4	0.6	0.3	1.3	1.0	KI, MS
2078				C_15_H_26_O	OS	0.2	0.3				KI, MS
2086	2127	1577	1576	spathulenol	OS	4.2	11.2	2.9	6.7	7.5	KI, MS
					Total	**93.5**	**81.7**	**91.0**	**91.1**	**88.4**	
Monoterpene Hydrocarbon (MH)					16	7.6	0	27.0	0.9	0	
Oxygenated Monoterpene (OM)					1	0	0	9.0	0	0	
Sesquiterpene Hydrocarbon (SH)					23	74.8	63.0	48.3	65.4	52.0	
Oxygenated Sesquiterpene (OS)					10	11.1	18.7	6.5	24.8	36.3	
Other (O)					1	0	0	0.2	0	0	
Total					**51.0**	**93.5**	**81.7**	**91.0**	**91.1**	**88.2**	

C. type = Compound type; MH = monoterpene hydrocarbon, OM = oxygenated monoterpene, SH = sesquiterpene hydrocarbon, SO = oxygenated sesquiterpene, O = Other; t = traces (<0.1%); KIa and KIb = linear retention index relative to n-alkanes on DB-Wax column or DB-5 column respectively; KI^1^ and KI^2^= literature linear retention index on DB-Wax column and DB-5 column respectively [48]; T.In. = *Trachymene incisa* subsp. *incisa*; I and II = typical glabrous and hairy variant respectively; I.M. = Identification method.

**Table 2 plants-10-00601-t002:** Location of the populations of *Trachymene incisa* subsp. *incisa* studied.

Sample	Voucher n°	Location	Yield
T.in.I_1_	LMC4175	Single National Park: (NSW, Australia) 20 km NW of Guyra (24 March 2007)	0.05%
T.in.I_2_	LMC4176		0.17%
T.in.I_3_	LMC4211		0.05%
T.in.II_4_	LMC4210		0.05%
T.in.II_5_	LMC4234	Dumaresq Dam: (NSW, Australia) Along southwestern side of Dumaresq Dam, c. 10 km NW of Armidale (14 April 2007)	0.1%

T.In = *Trachymene incisa* subsp. *incisa*; I and II = typical glabrous and hairy variant respectively.

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
