# Peer review of "The Essential Oil Composition of *Trachymene incisa* Rudge subsp. *incisa* Rudge from Australia [Author-notes fn1-plants-10-00601]"

_plants, 2021, doi:10.3390/plants10030601_

Round 1

Reviewer 1 Report

Dear authors, the work entitled "The essential oil composition of Trachymene incisa Rudge subsp. incisa from Australia" present significant aspects regarding the subject of this research. The work was well designed and the different sessions were well organised, however themanuscript requires further minor revision.

Introduction: page 2 line 59 "the development of an embryo, pregnancy.....". : delete comma and insert "in" before pregnancy;

The text lacks the bibliographical note [32] which is included in the session "References".

References:

References should be entered in order of year; for example: in the text it is stated on page 2 , line 48 [1, 3-9], References 3 to 9 are not in order of year , reference 3 is from 1993 while reference 6 is from 1992. Therefore, order all bibliographical notes by year in the References session.

Author Response

Thank you very much for all your comments. All of them have been considered:

1.- The sentence have been changed.

2.- Reference 32 have been included in the text [32-34].

3.- All the references have been arranged according to thier year when they appear together.

Reviewer 2 Report

The manuscript from Palá-Paúl describes a characterization of essential oil from Trachymene incisa using GC-MS.

The work is well presented and the instrumentation is state-of-the-art.

However, there a minor flaws surrounding this piece of work.

The article will benefit from a Venn diagram for the identified features. In my opinion, it will be more clear to the readers how many compounds are shared between the T. in I2 to II4.

It will be optimal if the author can provide a chromatogram of the essential oil as supplementary figures and I would also suggest to add a table with retention times and m/z for the identified metabolites.

I would recommend to use chemistry language and add information for the instrumention used. Particularly, mass range used during the analysis. Was the identification done based on full scan or MRM?

Author Response

Thank you very much for all your comments:

1.- We think that a Venn diagram is not necessary because the table has all the composition in detail. However, we have included the % of each type of compound in Figure 1.

2.- Supplementary figures with the Chromatogram of each population have been included, with the peak number. However, we think that the KI (linear retention index relative to n-alkanes) are the international parameters recognized, not the retention time. However we have included a supplementary table with the number of peak.

3.-  Some more text about how we ran the gc/ms to acquire data has been added in section 3.3.

Reviewer 3 Report

It is an interesting study written well and clearly, I recommend publishing it in its present form.

Suggestions: L 94 and 95 the sentence is unclear, please reformulate.

L 131 Figure 2, please include % (compound percentage) in all figures.

Author Response

Thank you very much for all your comments. All of them have been considered:

1.- The sentence has been rewritten.

2.- The % have been included in all the figures.